# Demonstration of Three True Random Number Generator Circuits Using Memristor Created Entropy and Commercial Off-the-Shelf Components

**DOI:** 10.3390/e23030371

**Published:** 2021-03-20

**Authors:** Scott Stoller, Kristy A. Campbell

**Affiliations:** Department of Electrical and Computer Engineering, Boise State University, Boise, ID 83725, USA; scott.stoller1@gmail.com

**Keywords:** true random number generator, memristor, entropy, multivibrator, oscillator

## Abstract

In this work, we build and test three memristor-based true random number generator (TRNG) circuits: two previously presented in the literature and one which is our own design. The functionality of each circuit is assessed using the National Institute of Standards and Technology (NIST) Statistical Test Suite (STS). The TRNG circuits were built using commercially available off-the-shelf parts, including the memristor. The results of this work confirm the usefulness of memristors for successful implementation of TRNG circuits, as well as the ease with which a TRNG can be built using simple circuit designs and off-the-shelf breadboard circuit components.

## 1. Introduction

Random numbers have a variety of uses in modern computing and information security, ranging from simple decision making in a video game, to encryption of secure documents and keeping banking transactions secure [1,2,3,4,5,6]. The security of data and communication channels is especially important today with the increase in connected devices throughout the world. Random number generators (RNGs) continue to be essential for keeping devices and communication channels secure.

RNGs fall in to two main types: pseudo random number generators (PRNGs) and true random number generators (TRNGs) [7]. PRNGs are often implemented as a linear feedback shift register (LFSR) or a linear congruential generator (LCG) [8] among others. One thing that separates PRNGs from TRNGs is the fact that all PRNGs are deterministic. That is, if the current state of the PRNG is known, then the future output of the PRNG can be predicted. A primary use of PRNGs is often scientific research and simulations (e.g., Monte Carlo) [6,9].

TRNGs do not generate random numbers based on a formula, but instead capture entropy from the environment to generate random numbers within hardware. Unlike PRNGs, the output from a TRNG is not deterministic and can never be guessed by knowing the previous outputs or current state of the generator. This is the primary reason that TRNGs are often used for securing data and communications channels.

TRNGs can be implemented in many ways. Examples of TRNGs include measuring the time between clicks on a Geiger counter [10], measuring frequencies or latencies of asynchronous events on a PC [11,12] and circuits comprised of oscillators where entropy is captured as jitter [3,4,5,7,13,14,15,16]. Even modern CPUs can have dedicated hardware on the application specific integrated circuit (ASIC) to capture entropy [17].

A recent approach in TRNG circuits has been to use a memristor device [18,19,20] to capture entropy for TRNG circuit designs [1,2,5,6,15,21,22,23]. Many of these designs use the memristors in a circuit that oscillates (e.g., a ring oscillator) or is driven by a pulse generator. In many cases, the memristor is simply used in place of a resistor in a more traditional oscillator circuit. Memristors offer a great platform for the simulation of stochastic events due to the nature of the filamentary memristor to be constituted of constantly rearranging atoms [20].

In this work, we implement two TRNG circuits recently presented in the literature [1,2] using commercially available off-the-shelf memristors [24] and parts [25]. We then put these circuits to the tests established by the National Institute of Standards and Technology (NIST) Statistical Test Suite (STS), which is used to assess the randomness of TRNGs [26]. Last, we build a memristor-based TRNG circuit of our own design and put it through the same tests in order to demonstrate the ease of implementation of memristor-based TRNGs and their accessibility to anyone using commercially available components. While there are other statistical test suites available to test the randomness of RNGs, the NIST STS had the most widespread use in our literature survey, having been used to assess RNGs in [1,2,5,6,7,8,14,15,16]. A brief description of each test in the NIST STS can be found in Appendix B.

## 2. Materials and Methods

### 2.1. Electrical Components and Measurements

Commercially available off-the-shelf parts were used to implement each of the circuits and the electrical tests. The memristor used was a discrete Tungsten Self Directed Channel (W-SDC) 16-pin dual in-line package (DIP) consisting of 8 memristor devices per package [24] The other circuit components were purchased at DigiKey [25] and the part numbers are listed in the schematic diagrams. All circuits were implemented on breadboards; however, the TRNG developed for this work was also implemented on a printed circuit board (PCB). The W-SDC memristor operates using a self-directed channel mechanism as described for the basic SDC memristor [18], except that the device structure is designed to provide a more continuous resistance change and operate using a mixture of phase-change and SDC mechanisms by including a thin layer of cosputtered W-Ge_2_Se_3_ between layers of Ge_2_Se_3_.

Electrical measurements were performed using a Digilent Analog Discovery 2 [27] with a breadboard breakout card which connected the AD2 to a breadboard. The AD2 was used to supply power to the circuit, generate the pulse train input and clock signals and collect the data using Digilent Waveforms software provided with the AD2. A stream of single bits was sampled on the rising or falling edge of the input data clock. One bit (or sample) is generated serially per clock cycle. Each bit represents the output of the TRNG for a single clock cycle. The bits were concatenated serially together into a CSV file and post-processed using a Perl script to convert them to a binary format. The Perl script used is provided in the Appendix A. Any additional details for electrical measurements specific to each circuit are described in the circuit description sections under 2.2 Circuits Tested. A total of 100 bitstreams each of length 1 million bits were tested for randomness for each circuit. The final binary bitstreams are a series of ones and zeroes that is serially written to a file by the Perl script that processes the CSV files. The Perl script used is included in the Appendix A.

Von Neumann debiasing [9] was used when post-processing data from the RNG circuits and is included in the Perl script generating the CSV files. Whitening (also known as debiasing) is a method of removing a bias in the output bitstream, for example if there are more zeroes than ones. Von Neumann or other types of debiasing algorithms are often used to debias the output of true random number generators [5,7,10,16,17]. It was found to be necessary to apply debiasing to the output of all three RNG circuits tested due to the bias in the output data from the TRNGs. Von Neumann’s debiasing scheme is simple to implement in either hardware or software. It provides a simple method of removing bias from a stream of bits without impacting the entropy of the bitstream. Debiasing is a technique that is commonly used in TRNGs. To describe this process, we use Figure 1 which shows a biased input bitstream that is debiased by Von Neumann whitening. Bits in the input bitstream are grouped into pairs. If the two bits are the same, then both bits are discarded. If the two bits are different, then only the first bit is kept. The resulting bitstream is the debiased output. In Figure 1, below, the first pair of binary bits are one, one. The bits are the same, so the sample is discarded. The next pair of bits, zero, one, are different. The first bit (zero) is kept and added to the final bitstream. This process is repeated until all pairs in the input bitstream have been processed.

### 2.2. Circuits Tested

Two previously described TRNGs using memristors were implemented in hardware with the W-SDC memristor device: Jiang’s [1] and Rai’s [2]. A third TRNG circuit tested was our own design, referred to as Student TRNG (S-TRNG). These circuits are shown in Figure 2.

### 2.3. Jiang’s TRNG

Jiang’s circuit, Figure 2a, was physically implemented in their reported work [1] using an Ag:SiO_2_-based diffusive memristor device. In Jiang’s design a pulse train is sent through a memristor that is placed in series with a resistor. Entropy is captured in the memristor device as a variability in the time it takes for the device to transition from a high resistance state to a low resistance state.

#### 2.3.1. Jiang’s Circuit Operation

The operation of Jiang’s TRNG is described in the timing diagram chart in Figure 3. When the output of the memristor-resistor circuit (Mem Out) is low (the memristor is a high resistance), below V_ref_, the comparator output (Clk_en_) is high. When Clk_en_ signal is high the counter is disabled. In the opposite case when the memristor is in a low resistance state, the output of the circuit is higher than V_ref_ and the Clk_en_ signal is low which enables the counter. This is a slight modification to Jiang’s original circuit which used an AND gate at the input of the counter to enable or disable the clock signal instead of the counter’s enable pin. The two circuits are functionally equivalent because the AND gate acts as an enable on the clock signal. If the enable input is a 0, then the output of the AND gate suppresses the clock and is always 0. If the enable input is a 1 then the output of the AND gate matches the input of the clock pin.

The time window region denoted by A in Figure 3 shows the time during which the random bit from the previous clock is sampled by the Digilent AD2. Line B denotes the rising edge of the pulse train. The counter is cleared during this time by a short pulse on the clear input. This is the time that the memristor will be programmed to a lower resistance state. At the time denoted by line C, the memristor changes from a high resistance to a low resistance. When the Mem Out voltage rises above V_ref_ the output of the comparator goes low. This enables the Clk_en_ on the counter, allowing it to count. At D, the pulse train voltage goes low, causing the Mem Out voltage to drop below V_ref_ and disable the counter, holding the output value of the MSB. The random bit generated in this case is held at “1”. The memristor will also be reset from a high to low state during this time. At E the cycle restarts. At F the transition of the memristor from high to low resistance occurs earlier than the previous cycle. The Clk_en_ signal goes high at G and the counter is disabled, this time holding its output at a “0” because a different number of clock cycles were counted than in the previous pulse train window.

#### 2.3.2. Breadboard Implementation and Measurements of Jiang’s Circuit

We implemented Jiang’s circuit on a breadboard using a W-SDC memristor, which is a different memristor technology from that used by Jiang. A 4 kHz pulse train frequency and a 50 MHz clock frequency (both generated by the Digilent AD2 Wavegen and pattern functions) were used during testing. V_ref_ was generated using a potentiometer between the Digilent AD2 V+ power supply and V- power supply. This allowed the V_ref_ voltage to be easily adjusted for varying pulse input voltage amplitudes. The least significant bit (LSB) output from the counter was sampled by a digital input on the AD2. Data was saved using a 20 kHz sample rate. A 22 kΩ resistor was used in series with the W-SDC memristor device, as seen in Figure 2a. The final binary output was post-processed using Von Neumann debiasing [9] and analyzed using the NIST STS application [28]. 100 samples of 1 million bits were tested. Elapsed time to collect these samples was approximately 40 h. Figure 4 shows an image of the breadboard implementation of Jiang’s TRNG circuit.

### 2.4. Rai’s TRNG

The second TRNG implemented in this paper, proposed by Rai et al., is a design that takes inspiration from a common dual inverter oscillator TRNG design [2]. Rai’s design has not previously been physically implemented prior to our work herein. Instead, the design was simulated by Rai using the TiO_2_ memristor model described in [29]. Entropy is captured as the time that it takes for a conductive channel to form in the memristor device, or the time that it takes the device to transition from high resistance to low resistance.

#### 2.4.1. Rai’s Circuit Operation

Rai’s TRNG circuit consists of two inverter delay paths, with a memristor in series with the inverter devices in each delay path [2]. The outputs of the two inverter strings are both sampled and the output of the TRNG is based on which output switches first. If the D_first_ output switches first the output is 0. If the D_second_ output switches first, then the output is a 1.

In Rai’s circuit simulation a latch was used. In our circuit implementation (Figure 2b) we used a comparator to act as the latch since it has a latching feature. We implemented the circuit with one delay path connected to the input of the comparator and the other delay path connected to the latch input on the comparator. If the first delay path is faster, the output of the comparator is driven high before the output is latched by the second delay path. If the second delay path is faster, the output of the comparator is latched at a low output.

#### 2.4.2. Breadboard Implementation and Measurements of Rai’s Circuit

The breadboarded circuit for Rai’s TRNG design is shown in Figure 5. TI CD40106BE inverter chips were used along with the W-based SDC memristors to create the inverter-memristor delay path chains. The circuit input was driven by a square wave clock generated from the Digilent AD2 Wavegen. A frequency of 2 kHz was used for the pulse train clock input. An Analog Devices AD8561 comparator was used to capture which device switched first by using the latch input on the comparator. V_ref_ was generated by an analog output from the Digilent AD2. 100 samples of 1 million bits were tested. The lapsed time to collect this data was approximately 80 h.

### 2.5. S-TRNG

A new memristor-based TNRG circuit, referred to as the student TRNG, or S-TRNG, was created, implemented and characterized for comparison to the previous two circuits. The circuit diagram for this design is shown in Figure 2c. The core concept of the S-TRNG circuit is similar to many other TRNGs [3,4,5,7,13,14,15,16]. The circuit consists of two oscillators. One oscillator runs at a fast speed and the other oscillator runs at a slow speed. The slow oscillator acts as a clock to sample data from the fast oscillator. Entropy is captured in the slow oscillator as jitter or variability in the period of the oscillator. In the case of the S-TRNG two slow oscillators are used and the outputs are XOR’d together to increase the frequency of the resulting slow clock. This allowed us to collect data at a faster rate.

#### 2.5.1. S-TRNG Circuit Operation

The S-TRNG circuit consists of two memristor multivibrator oscillators feeding into an XOR gate. The output of the XOR gate feeds into the latch input of a comparator. This acts as a slow oscillator that samples the output of a much faster oscillator. A third multivibrator circuit with a resistor (not memristor) is used to generate the fast oscillations. The output of this circuit is fed into the input of the comparator. In this way, the fast oscillator is sampled by the clock generated by the slow oscillators.

The multivibrator circuit consists of an op-amp with a voltage divider from the output of the op-amp connected to the noninverting input of the op-amp. A memristor that charges a capacitor is connected form the output to the inverting input of the op-amp. The output of the op-amp oscillates between the positive supply voltage (V_CC_) and the negative supply voltage (−V_CC_). When the output is at V_CC_, the voltage at the noninverting input is 1/2 V_CC_ due to the voltage divider. The voltage at the inverting input will be charged from −1/2 V_CC_ to 1/2 V_CC_. Once he inverting input reaches a voltage of 1/2V_CC_ (the same voltage at the noninverting input), the output of the op-amp will switch to −V_CC_. The voltage at the noninverting input will switch to −1/2 V_CC_ and the capacitor will begin discharging from a voltage of 1/2 V_CC_ to a voltage of −1/2 V_CC_. Once a voltage of −1/2 V_CC_ is reached at the inverting input, the cycle will start over again.

The oscillation period of the multivibrator circuit is the total time it takes to charge the capacitor from −1/2 V_CC_ to 1/2 V_CC_ and then discharge back down to −1/2 V_CC_. The oscillation period of the multivibrator circuit can be derived for a generic memristor device with a high resistance of R_H_, a low resistance of R_L_ and a time to switch of R_LH_ by examining two time windows of operation. The first time window is the charging of the capacitor while the memristor is in a low resistance state until it switches to a high resistance state. This is modelled by Equation (1) to calculates V_LH_, the voltage at which the memristor device switches from a high resistance to low resistance state.
(1)$$V_{LH}=V_{CC}-1.5V_{CC}e^{-\frac{T_{LH}}{R_{L}C}}$$

The second time window is shown in Equation (2) calculates T_HLC_ as the time from when the device switches low resistance to high resistance state until the capacitor is fully charged.
(2)$$T_{LHC}=-R_{H}C\ln\left(\frac{1}{2}\frac{V_{CC}}{V_{CC}-V_{LH}}\right)$$

Equation (3) is the sum of the time to switch from high resistance to low resistance and the remaining time to charge the capacitor.
(3)$$T_{C}=T_{LH}+\;T_{LHC}$$

Equations (1) and (2) can be simply flipped to model the discharge of the capacitor. Equation (3) is easily adapted for the discharge portion of the multivibrator output cycle to get Equation (4).
(4)$$T_{D}=T_{HL}+\;T_{HLD}$$

Equation (5) is simply the sum of Equations (3) and (4) and gives the period of one oscillation of the multivibrator. Equation (5) shows the time of a single oscillation period of the multivibrator with a memristor.
(5)$$T_{OSC}=T_{C}+\;T_{D}=T_{LH}+\;T_{LHC}+T_{HL}+\;T_{HLD}$$

As the switching delay of the memristor changes with each cycle of the memristor in the multivibrator circuit, entropy is captured by the TRNG as a random stream of bits. It must be noted that it is essential to choose a capacitor value that is large enough such that it takes longer to charge or discharge the capacitor than it does for the memristor to switch from high resistance state to low resistance state or vice versa.

Figure 6 show histograms of the variability in a measurement of the clock period of the multivibrator oscillator design implemented with a memristor (top) or a resistor (bottom). The period is measured as the time from one rising edge to the next rising edge of the output of the oscillator. The variability of the period of the oscillator is significantly greater when the memristor device is used in the multivibrator circuit than when the resistor is used. Figure 7 shows an example clock period for the memristor-based oscillator (with high variability of clock period) and the resistor-based oscillator (with low variability of clock period).

#### 2.5.2. Breadboard Implementation and Measurements of S-TRNG Circuit

The circuit was initially prototyped on a breadboard (Figure 8a) and then designed and tested on a PCB (Figure 8b). Similar responses were measured using the NIST STS tests for the two different implementations. Therefore, final testing was performed using the PCB. As with the Jiang and Rai circuits, 100 bitstreams each of length 1 million bits were tested with the PCB S-TRNG circuit. One bit of data is collected for each clock cycle of the slow oscillator of the TRNG. In contrast to the Jiang and Rai TRNGs, the S-TRNG is self-clocked by the slow oscillator in the circuit. In this case it is necessary to asynchronously sample the output of the S-TRNG with the Digilent AD2. It was also necessary to oversample by a factor of more than 2× due to the variability in the clock period of the memristor-based slow oscillator. Both Jiang and Rai’s circuits were clocked by the Digilent AD2 allowing use of the pulse train clock as the data collection clock for those TRNGs.

Capacitor values of 1 nF were used for the slow oscillators with the W-SDC devices resulting in an oscillation frequency that could range from about 500 Hz to 5 kHz, depending on the memristor device state. Resistor and capacitor values of 2.2 kΩ and 100 pF were used for the fast oscillator circuit, resulting in an oscillation frequency of 600 kHz. Due to the significant amount of variability in the output clock frequency of the slow oscillator it was necessary to oversample by a factor more than 2x. A clock frequency of 40 kHz was used to sample the output of the circuit. With each slow oscillator operating at a frequency of approximately 4 kHz it took about 25 h to complete the data collection.

## 3. Results

The NIST statistical test suite [26] briefly described in Appendix B was used to assess the randomness of all three TRNGs implemented. All NIST tests were run with the default settings and parameters listed in Appendix B. Data for all TRNGs were post-processed with Von Neumann whitening. The results of these tests are summarized in Table 1 where each NIST test is listed along with the passing rate for that test for each circuit. Two additional table columns are listed to show a comparison of the prior published test results for the Jiang circuit using a different memristor type and the published simulation results for the Rai against the measurements made in this work. In order to be considered passing, a proportion of at least 96% of the 100 bitstreams for each test must pass.

Jiang reported passing the STS tests (first data column in Table 1) using the Ag:SiO_2_-based diffusive memristor device with >96%. The hardware implementation (second data column in Table 1) using the W-SDC memristor also shows all tests passing. In order to pass the frequency test during the W-SDC circuit implementation in our work it was necessary to apply debiasing to the output of the TRNG. Without debiasing Jiang’s TRNG circuit had a pass rate is 0% for the STS frequency test. Jiang’s circuit performed slightly worse in our implementation than in [1]. This could be due to a variety of factors, most likely the fact that a different type of memristor device was used for our testing.

Rai’s simulated results using the TiO_2_ model [29] are given in the third results column in Table 1. Only five NIST tests were simulated in Rai’s report. However, the hardware implementation of Rai’s TRNG using the W-SDC memristor (Table 1, fourth data column) shows that the Rai TRNG passes every NIST test with superior performance to any other TRNG tested in our study. However, without debiasing our hardware implementation of Rai’s TRNG has a pass rate of 0% for the STS frequency test, similarly to the result obtained for the Jiang circuit.

The last data column in Table 1 shows the S-TRNG test results 13 of the 15 NIST STS tests are passing for this circuit. The Runs test failed with an 82% pass rate and the approximate entropy test fails with a 94% pass rate. The runs test is a measure of the number of runs within a sequence. A run is defined as a repetition of the same value within the sequence. For example, a run of zeros with length 4 would be “0000”. If the number of runs of each length does not match the expected distribution of runs within a random sequence, then the runs test fails. The approximate entropy test analyzes patterns within the data, specifically looking at sequences of length m and m+1 bits. If the distribution of patterns within the random sequence does not match the expected value, then the approximate entropy test fails. A thorough examination of the circuit for sources of potential noise could eliminate certain deterministic behavior and improve the pass rate of the circuit for these two tests.

Throughput of the TRNGs tested varied by design. Throughput is a measure of how quickly each TRNG is able to generate random data. All TRNGs ran until 100 million bits were collected after Von Neumann whitening. The S-TRNG was the quickest at 25 h (1111 bits per second). Jiang was the next highest throughput at 40 h (694 bits per second). Rai required 80 h to collect data (347 bits per second). It should be noted that the throughput of the TRNGs depends greatly on the devices and circuit component values used in the case of the S-TRNG. The oscillation frequency of the S-TRNG can be affected by both the W-SDC device variability, as well as the chosen capacitor value in the circuit. Similarly, both Jiang and Rai TRNGs throughput are affected by the frequency at which the pulse train runs. This is limited by the time that it takes for the memristor devices to switch states.

## 4. Discussion

Three TRNG circuits were physically implemented and tested using W-SDC memristors. The Jiang and Rai memristor-based TRNG circuits passed all NIST STS tests for true randomness. The student TRNG (S-TRNG) passed 13 out of 15 of the NIST STS tests. There are several pros and cons of each TRNG analyzed in this work. The S-TRNG circuit is more complex than both Rai and Jiang and does not generate random numbers as well as Rai or Jiang. However, the S-TRNG design is self-clocked; that is, an external clock signal does not need to be provided to the circuit. This can be an advantage in some cases. In addition, it was found during testing Rai’s TRNG that some fine-tuning was required in order to get the delay chain time delays close to each other in order to produce meaningful output. This could be accomplished a couple different ways, by either swapping out memristor devices until two devices were found that gave similar delays or adding resistance in series with the memristor or adding small amounts of capacitance at the output of the memristor. In addition, it was found that the delicate balance of delay paths can shift throughout the course of testing. This is one area where simulation alone is not sufficient to prove the randomness of the TRNG. The S-TRNG required no tweaking in this manner. All TRNGs required whitening of the output in order to pass the frequency test. It was found during initial development of the S-TRNG circuit that the W-SDC memristor in place of a resistor in the multivibrator circuit resulted in a significant increase in the amount of jitter produced by the circuit. This property was exploited to significantly increase amount of entropy captured by the circuit. It was also found that it can be useful to implement a counter after the XOR gate and before the input to the latch to essentially divide the slow clock to an even slower frequency. This effect allows for more entropy to be captured because the jitter from multiple slow clock cycles now contributes noise to the sampling of the fast clock. In addition, the frequency of the slow clock can be more easily adjusted by selecting a different output from the binary counter IC.

Throughout the extensive development and testing of the S-TRNG circuit it was found that one of the primary factors that can lead to non-random output from the TRNG is supply noise. Power supply ripple can be observed due to the instantaneous high current observed when the multivibrator circuit switches from one mono-stable state to the other. This supply ripple from one oscillator switching can lead to other oscillators sharing the same supply switching at the same time. In order to improve the power supply isolation between the slow oscillators and the fast oscillator, the voltage generator outputs on the Digilent AD2 were used to power the fast oscillator while the normal power supply outputs were used to supply the slow oscillators. This helps to ensure that there is no correlation between the fast and slow oscillators due to power supply noise injected by one oscillator or the other.

## 5. Conclusions

Memristors have been physically implemented in circuits to produce true random number generation through entropy capture. Two circuits described in the literature were implemented (Jiang [1] and Rai [2]) and one additional circuit was design and implemented in this work. The goal of this work was to demonstrate that true random number generator circuits are readily achievable with off-the-shelf components and circuit designs at many levels of sophistication.

## Figures and Tables

**Figure 1 entropy-23-00371-f001:**
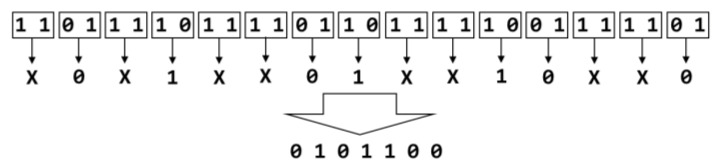
Example of Von Neumann whitening.

**Figure 2 entropy-23-00371-f002:**
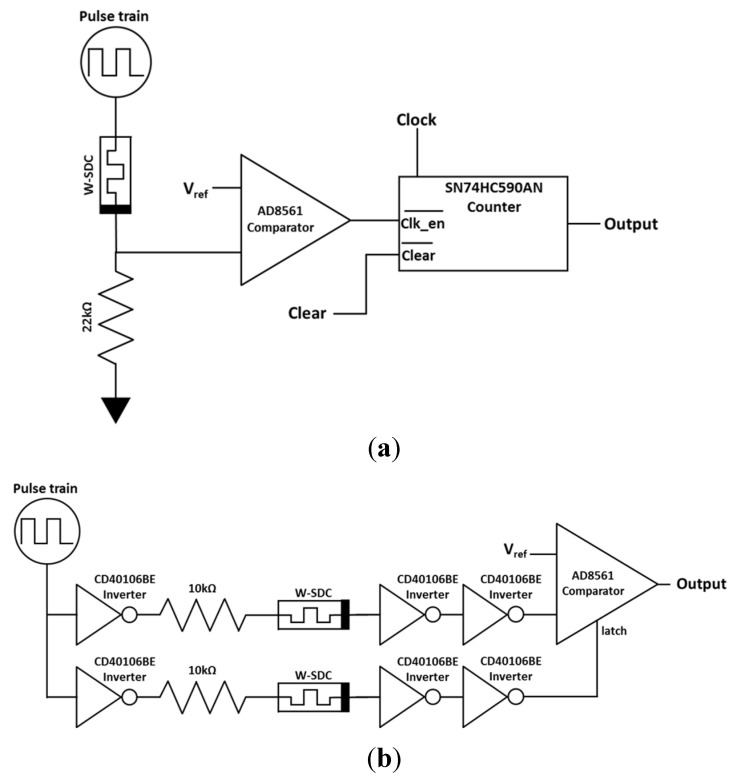
Circuits tested. (**a**) Jiang’s circuit modified for testing [1]; (**b**) Rai’s circuit adapted for testing [2]; (**c**) Our design, student-true random number generator (S-TRNG).

**Figure 3 entropy-23-00371-f003:**
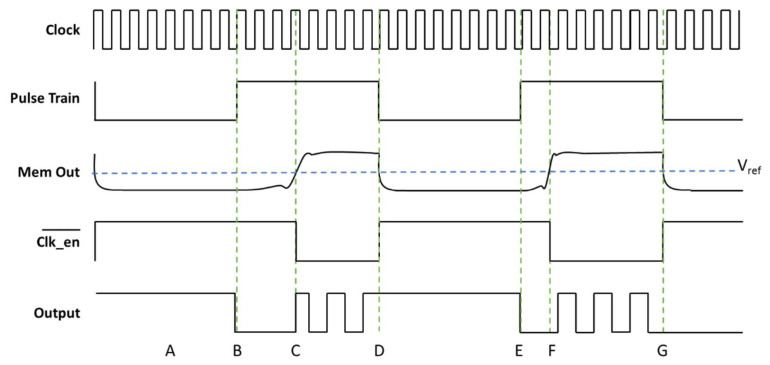
Timing diagram explanation of Jiang’s true random number generator circuit.

**Figure 4 entropy-23-00371-f004:**
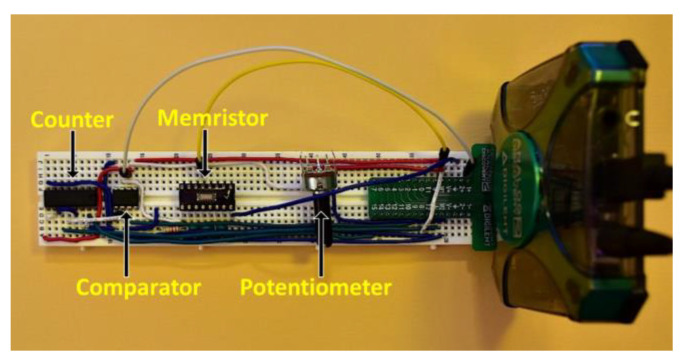
Implementation of Jiang’s design on a breadboard. The circuit is connected to the Digilent AD2 on the right.

**Figure 5 entropy-23-00371-f005:**
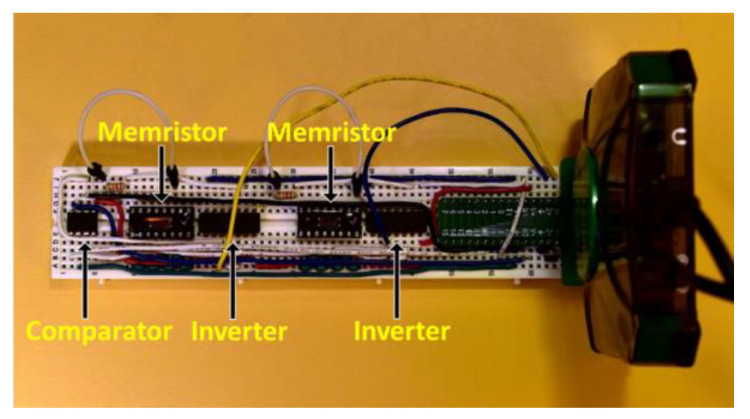
Breadboard implementation of Rai’s true random number generator circuit. The circuit is connected to the Digilent AD2 on the right.

**Figure 6 entropy-23-00371-f006:**
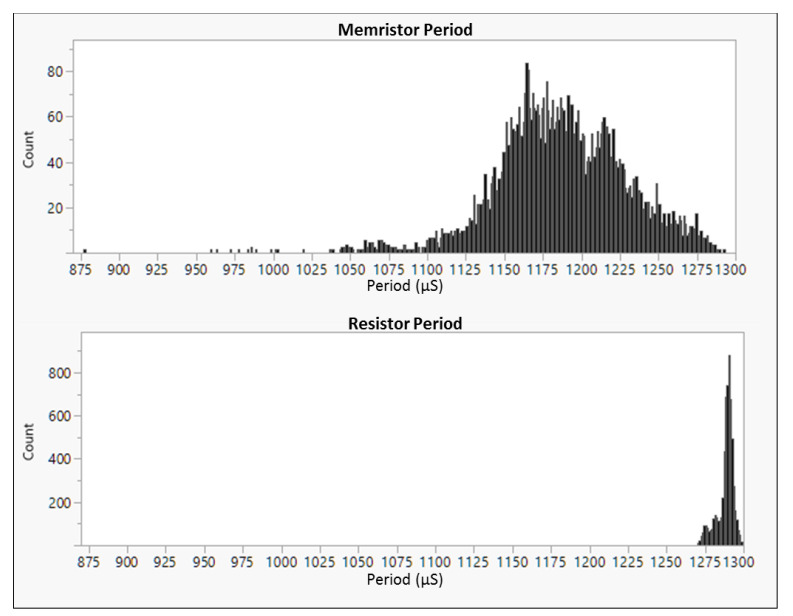
Histograms of the entropy captured as a slow clock sampling a fast clock in a dual oscillator type random number generator used in the S-TRNG circuit. Top graph: with a memristor. Bottom graph: with only a resistor. A total of 6000 clock periods were sampled for each oscillator type.

**Figure 7 entropy-23-00371-f007:**
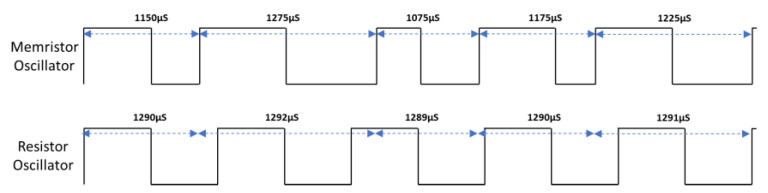
Example clock periods for memristor-based and resistor-based oscillators (variability emphasized in this example).

**Figure 8 entropy-23-00371-f008:**
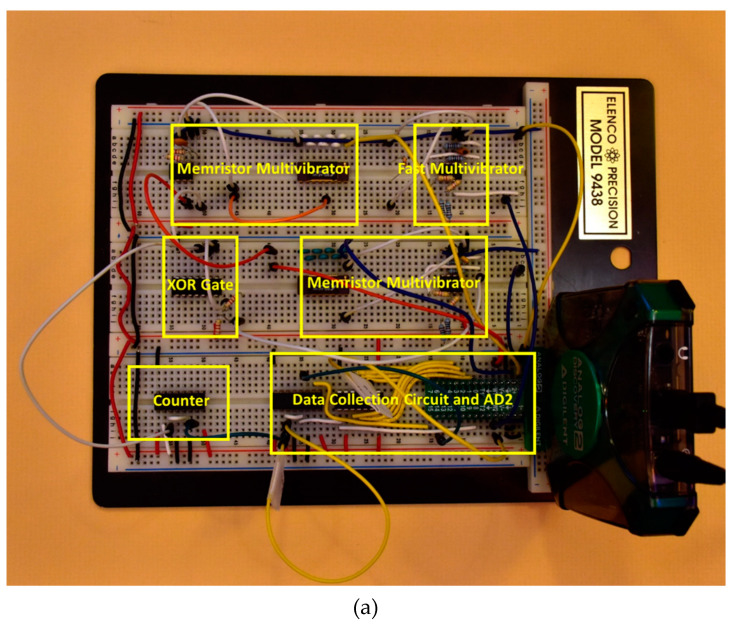
(**a**) Breadboard implementation; (**b**) Printed circuit board (PCB) implementation of the S-TRNG circuit.

**Table 1 entropy-23-00371-t001:** Comparison of all TRNG measurement results for the 15 STS tests and additional sequence, debiasing and implementation comparison for each dataset. Pass rates are shown for each STS test. Bolded pass rates are considered failing (less than 96%).

NIST STS Test	Jiang TRNG (from [1])	Jiang TRNG	Rai TRNG (from [2])	Rai TRNG	S-TRNG
Frequency	97%	99%	100%	100%	98%
Block Frequency	99%	99%	-	100%	98%
Cumulative Sums	97%	99%	100%	100%	97%
Runs	99%	98%	100%	100%	82%
Longest Run	100%	99%	-	100%	100%
Rank	100%	96%	-	100%	98%
FFT	99%	99%	100%	100%	97%
Non Overlapping Template	98%	99%	-	99%	99%
Overlapping Template	99%	98%	-	100%	98%
Universal	100%	99%	-	100%	100%
Approximate Entropy	99%	99%	100%	100%	94%
Random Excursions	98%	98%	-	96%	98%
Random Excursions Variant	99%	99%	-	98%	99%
Serial	100%	99%	-	96%	98%
Linear Complexity	100%	99%	-	100%	100%
Sequence Length	1,000,000	1,000,000	5000	1,000,000	1,000,000
Sequences Tested	76	100	100	100	100
Debiasing applied	No	Yes	No	Yes	Yes
Circuit Implementation	Hardware	Hardware	SPICE	Hardware	Hardware
Memristor Device	Ag:SiO_2_	W-SDC	Model for TiO_2_ [28]	W-SDC	W-SDC

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
