# Peer review of "Demonstration of Three True Random Number Generator Circuits Using Memristor Created Entropy and Commercial Off-the-Shelf Components"

_entropy, 2021, doi:10.3390/e23030371_

Round 1

Reviewer 1 Report

The work of Stoller and Campbell reports on the use of memristors for the implementation of TRNG. The authors discuss to existing configurations while proposing a new one, all the configurations are tested experimentally after describing their electronic schematics. The results indicate a solid pathway for using memristors in the space of TRNG with simple circuit design implementation. The paper is well written, and it can be considered further for publication.

As a minor comment the authors, they are invited to add in their introduction a short note on the intrinsic reason why memristors offer a great platform for the simulation of stochastic events, i.e., the nature of filamentary memristor to be constituted of constantly rearranging atoms. In this respect the authors are invited to add existing experimental works to show this [Yang, Y.; Gao, P.; Li, L.; Pan, X.; Tappertzhofen, S.; Choi, S.; Waser, R.; Valov, I.; Lu, W. D. Electrochemical Dynamics of Nanoscale Metallic Inclusions in Dielectrics. Nat. Commun. 2014, 5 (May), 4232. https://doi.org/10.1038/ncomms5232.] and [https://doi.org/10.1021/acs.nanolett.5b03078]   

Reviewer 2 Report

This paper well describes the implementation results of three memristor-based TRNG circuits. Here are some suggestions I would like to make to improve the manuscript.

  1. The comparison regarding the area and power consumption aspect is highly recommended. 
  2. It is recommended to include analysis and explanation of the phenomenon in which the bitstreams generated from S-TRNG fail in the "run" and "approximate entropy" tests.
  3. A quantitative comparison of throughput of the TRNG bitstream can be included.

Reviewer 3 Report

This article simply describes the evaluation of 3 TRNG implementations (two of which already exist).

I have not been able to understand the main contribution:
- is it the fact that it is simple to implant TRNG?
- is it the student RNG?
In any case, it does not seem to me sufficient to be published as is in a newspaper like Entropy.

Minor remarks : 
- In the introduction, the authors state that PRNGs cannot be used for cryptography.
However, this is done every time using cryptographically secure PRNGs to generate encryption keys (AES). I think that algorithms referenced in the
https://en.wikipedia.org/wiki/Cryptographically_secure_pseudorandom_number_generator
should be studied to nuance the discourse.
The PRNGs referenced in the introduction based on LCG and LFSR are no longer used today, except for simulation. However, their linearity properties are so bad that they should be used in a context that is not based on them.

 - "we buil two TRNG" line 51 is not clear. Reading the rest of this article, we understand that it is the implementation on physical devices that is carried out.

- VonNeumann's Debiasing is an algorithm and therefore reproducible. This seems to be in contradiction with the objective of TRNGs which are based on physical devices. the author should detail how they handle this fact.  

- line 114: "the two circuits are functionally equivalent". This statement is claimed and should be proven. 

- I think that the 3 physical devices should be presented separately, compared physically, compared their flow rates and finally compared their statistical properties.

- I don't understand the interest of the 3 pictures. 

- In Table 1, the two columns for Jiang TRNG give different results: your evaluation via the NIST test consistently gives worse results than those presented in the review. This should be explained.

- In Table 1, the column for S-TRNG clearly gives poor results for Runs. How do the authors explain this. How can this be corrected?

- Finally, more stringent test like TESTUO1 or PractRand should be considered.

Round 2

Reviewer 3 Report

The authors have addressed the minor remarks I made, and I thank them for that.

The main contribution of the work remains the implementation of TRNG from memristor and the evaluation of these implementations using the NIST test. If it is of practical interest, I think the contribution is small: 
- the implementations have been simple to perform according to the authors.
- The limitations of the NIST test are not sufficiently explained.

I find these contributions a bit weak for a journal like entropy.

Author Response

Thank you for your feedback.